# Association Between Healthy Dietary Patterns and Chronic Kidney Disease in Patients with Diabetes: Findings from Korean National Health and Nutrition Examination Survey 2019–2021

**DOI:** 10.3390/nu17091600

**Published:** 2025-05-07

**Authors:** Minsang Kim, Jung Hun Koh, Jeong Min Cho, Semin Cho, Soojin Lee, Hyuk Huh, Seong Geun Kim, Sehyun Jung, Eunjeong Kang, Sehoon Park, Jin Hyuk Paek, Woo Yeong Park, Kyubok Jin, Seungyeup Han, Kwon Wook Joo, Kyungdo Han, Dong Ki Kim, Yaerim Kim

**Affiliations:** 1Department of Internal Medicine, Seoul National University Hospital, Seoul 03080, Republic of Korea; kminsang71@gmail.com (M.K.); kohjhsd@gmail.com (J.H.K.); fomalhaut23@naver.com (E.K.); mailofsehoon@gmail.com (S.P.); junephro@gmail.com (K.W.J.); dkkim73@gmail.com (D.K.K.); 2Department of Internal Medicine, Chung-Ang University Gwangmyeong Hospital, Gwangmyeong 14353, Republic of Korea; als6494@naver.com (J.M.C.); jseminy@gmail.com (S.C.); 3Department of Internal Medicine, Uijeongbu Eulji University Medical Center, Uijeongbu 11759, Republic of Korea; creektale@hanmail.net; 4Department of Internal Medicine, Inje University Busan Paik Hospital, Busan 47392, Republic of Korea; hh820616@hanmail.net; 5Department of Internal Medicine, Inje University Sanggye Paik Hospital, Seoul 01757, Republic of Korea; kimsob88@gmail.com; 6Department of Internal Medicine, Gyeongsang National University Hospital, Jinju 52727, Republic of Korea; holyhyun@naver.com; 7Transplantation Center, Seoul National University Hospital, Seoul 03080, Republic of Korea; 8Department of Internal Medicine, Keimyung University School of Medicine, Daegu 42601, Republic of Koreawy-my@hanmail.net (W.Y.P.); mdjin922@gmail.com (K.J.); hansy@dsmc.or.kr (S.H.); 9Department of Internal Medicine, Seoul National University College of Medicine, Seoul 03080, Republic of Korea; 10Department of Statistics and Actuarial Science, Soongsil University, Seoul 06978, Republic of Korea

**Keywords:** chronic kidney disease, diabetes mellitus, healthy dietary pattern, healthy eating index

## Abstract

**Background/Objectives**: Although a healthy dietary pattern is a modifiable lifestyle factor in the prevention of chronic kidney disease (CKD), studies that investigate the association between a healthy diet and prevalent CKD in patients with diabetes, using the Korean Healthy Eating Index (KHEI), are lacking. **Methods**: This cross-sectional study included 1991 patients with diabetes from the eighth Korean National Health and Nutrition Examination Survey 2019–2021. A higher KHEI indicated healthier eating habits. CKD was defined as an estimated glomerular filtration rate < 60 mL/min/1.73 m^2^ or urine albumin–creatinine ratio ≥ 30 mg/g. The risk of prevalent CKD was evaluated according to the median KHEI value using logistic regression analysis adjusted for various clinicodemographic characteristics. Each KHEI component score was compared between those with and those without CKD, using the Student’s *t*-test. **Results**: Participants with a higher KHEI were older, with higher proportions of women, non-smokers, and non-alcoholics. A higher KHEI was significantly associated with a lower risk of prevalent CKD (adjusted odds ratio [aOR], 0.73 [0.58–0.93]). Subgroup analysis revealed stronger associations in those without hypertension status (aOR, 0.57 [0.37–0.87]) with at least high school education (aOR, 0.56 [0.38–0.81]). Moreover, patients with diabetes and CKD had significantly lower KHEI, particularly in the adequacy category components, including breakfast consumption, total fruit intake, and dairy product intake. **Conclusions**: A healthier dietary pattern was associated with a lower risk of prevalent CKD in patients with diabetes. Dietary intervention, which recommends the intake of breakfast, fruits, and dairy products, may be an effective strategy for CKD prevention.

## 1. Introduction

Diabetes mellitus (DM) is a significant public health concern in Korea, with a prevalence of 12.5% among adults. The medical expenditures pertaining to diabetes have exhibited a consistent upward trend, reaching a total medical cost of 2.5 billion United States dollars in 2022 [1]. Moreover, DM increases the risk of chronic kidney disease (CKD) development and aggravation [2,3]. In Korea, 25.4% of patients diagnosed with DM also have diabetic kidney disease [4]. Given the increased risk of comorbidities and mortality associated with CKD in patients with DM [5,6], prevention and management of CKD in patients with diabetes has become a key objective of their overall management. Therefore, it is essential to identify the modifiable risk factors associated with CKD in patients with diabetes.

Diet is a modifiable lifestyle factor, and a healthier diet may help mitigate the risk of CKD [7,8]. Since a meal is consumed in whole foods rather than a single nutrient, a diet’s quality is assessed using certain indices of dietary patterns [9], such as the Healthy Eating Index (HEI), which was derived from the American dietary guidelines [10,11,12]. In previous studies, a healthy dietary pattern, as assessed by the HEI, was associated with a lower risk of the incidence and progression of CKD in both the general population [13,14] and patients with diabetes [15,16]. However, dietary patterns and diet quality are different among different ethnic groups [17].

The fundamental components of the traditional Korean diet consist of steamed rice, a variety of vegetables in raw, pickled, and steamed forms, and protein sources, including fish, meat, and soy-based products. Although the Korean diet is gradually adopting more Western patterns, which are characterized by a high intake of animal products and fat, Koreans continue to exhibit a high intake of carbohydrates, with a relatively low proportion of energy derived from protein and fat [18]. Consequently, in Korea, the Korean Healthy Eating Index (KHEI) was developed based on the dietary guidelines for Koreans [19,20]. However, to date, no studies have investigated the association between the KHEI and CKD risk in both the general population and patients with diabetes.

Therefore, we aimed to investigate the association between healthy dietary patterns, as assessed by the KHEI, and the risk of prevalent CKD in patients with diabetes and identify the differences in dietary patterns between patients with diabetes with CKD and those without CKD. We hypothesized that a healthy diet is associated with a lower risk of prevalent CKD in patients with diabetes.

## 2. Materials and Methods

### 2.1. Study Setting and Population

The KNHANES is a nationwide, population-based, cross-sectional survey that was established in 1998 and conducted annually to examine the health behaviors, prevalent chronic diseases, and dietary habits of the Korean population; it is administered by the Korea Centers for Disease Control and Prevention [21]. This cross-sectional study used data from the eighth KNHANES, which was conducted between 2019 and 2021. A two-stage stratified cluster sampling method was employed, with districts and households as the primary and secondary sampling units, respectively. Within each sampled district, a random selection of households was made, with the exclusion of collective living facilities (e.g., nursing homes, military bases, or prisons) and foreign households. The survey was conducted with all household members aged one year or older. A total of 22,559 individuals who participated in the KNHANES between 2019 and 2021 were included in this study. Individuals aged <19 years; those with missing data on KHEI, serum creatinine and urine albumin levels, and other essential covariates; and those without DM were excluded.

### 2.2. Study Outcomes and Ascertainment of CKD

The primary outcome was the risk of prevalent CKD defined as an estimated glomerular filtration rate (eGFR) < 60 mL/min/1.73 m^2^ or urine albumin–creatinine ratio (UACR) ≥ 30 mg/g; eGFR was calculated with the Modification of Diet in Renal Disease equation based on the Jaffe serum creatinine value and UACR was derived by calculating the ratio of urine albumin concentration to urine creatinine concentration.

### 2.3. Assessment of Healthy Dietary Pattern and KHEI

Overall diet quality was assessed using the KHEI, which was calculated using a nutrition survey comprising an assessment of dietary habits and a 1-day 24-h recall. Trained dietitians interviewed the participants at home during the nutrition survey. The KHEI comprises three categories: adequacy, moderation, and balance. The adequacy category evaluates adequate intake of the recommended foods and includes eight components: (1) frequency of breakfast consumption; (2) mixed grains intake; (3) total fruits intake; (4) fresh fruits intake; (5) total vegetable intake; (6) vegetable intake, except kimchi and pickled vegetables; (7) meat, fish, eggs, and beans intake; (8) milk and dairy products intake. The moderation category evaluates dietary restraint and includes three components: (1) the percentage of energy from saturated fatty acids; (2) sodium intake; (3) the percentage of energy from sweets and beverages. The balance category evaluates macronutrient and energy intake balance and includes three components: (1) the percentage of energy from carbohydrate; (2) percentage of energy from fat; (3) energy intake. Thus, the KHEI consists of 14 components with a total score of 100 [19]. The minimum and maximum scores for each component were determined based on the 2015 Dietary Reference Intakes for Koreans, dietary guidelines, and the observed distribution of food intake in Korea. If the intake of food or nutrients was within the minimum and maximum ranges, proportional scores were calculated according to the quantity of intake. Detailed scoring methods for the KHEI are provided in Appendix A. Although further validation studies of the KHEI are warranted, it is commonly used in research utilizing KNHANES data, as the KHEI was developed to evaluate the overall diet quality of Korean adults based on the National Dietary Guidelines for Koreans and the 2015 Dietary Reference Intakes for Koreans [22].

### 2.4. Data Collection and Measurements

In the KNHANES, sociodemographic information such as age, sex, and socioeconomic status (SES), including education level, income status, number of household members, residential area, occupation, and marital status, was collected. Information on lifestyle factors, including smoking status, alcohol consumption, and physical activity, was collected using self-reported questionnaires. Smoking status was classified as non-smoker, ex-smoker, or current smoker. Alcohol consumption status was categorized as non- (less than once per month), mild-to-moderate, or heavy (more than four times per week) consumption. Regular physical activity was defined as 20 min of vigorous-intensity physical activity engaged ≥3 days per week or 30 min of moderate-intensity physical activity engaged ≥5 days per week. Anthropometric measurements, including height, weight, waist circumference, and systolic and diastolic blood pressure, were obtained. Body mass index (BMI) was calculated as weight (kg) divided by height squared (m^2^) and expressed in kg/m^2^. Laboratory measurements, including fasting plasma glucose (FPG), glycated hemoglobin (HbA1c), and lipid profiles, were also obtained. Diabetes was defined as an FPG ≥ 126 mg/dL, HbA1c ≥ 6.5%, the use of diabetes medication or insulin, or a diagnosis by a doctor. Hypertension was defined as a systolic blood pressure ≥ 140 mmHg, a diastolic blood pressure ≥ 90 mmHg, the use of antihypertensive agents, or a diagnosis by a doctor. Dyslipidemia was defined as a total cholesterol level of >240 mg/dL or the use of dyslipidemia medication.

### 2.5. Statistical Analysis

Continuous variables are reported as mean ± standard error, and categorical variables are reported as numbers (percentages). For baseline characteristics, the participants were divided into two groups based on the median value of the KHEI in the total study population: low and high. We performed logistic regression analysis to assess the risk of prevalent CKD by setting the low-KHEI group as the reference group. Additionally, we performed a logistic regression analysis by setting the KHEI as a continuous variable. Considering potential confounding effects, a multivariable model adjusted for age, sex, body mass index (BMI), history of hypertension and dyslipidemia, education level, income status, occupation status, smoking, alcohol consumption, and physical activity was constructed. A restricted cubic spline curve was plotted to describe the risk of prevalent CKD as the KHEI was continuously altered. Subgroup analyses stratified by age (65 years), sex, presence of hypertension, and SES were conducted to evaluate the risk of prevalent CKD in diverse subgroups. The participants were categorized into two groups according to the presence of CKD, and the mean scores of the total KHEI and each KHEI component were then compared using the Student’s *t*-test. Furthermore, the mean KHEI scores were adjusted for the same covariates included in the multivariate model used for the logistic regression analysis. All statistical analyses were performed using SAS (version 9.4, SAS Institute, Cary, NC, USA), and two-sided *p* values < 0.05 were considered statistically significant.

## 3. Results

### 3.1. Baseline Characteristics

A total of 1991 participants were included in the cross-sectional analysis (Figure 1). The baseline characteristics of the study participants divided by the median KHEI value are presented in Table 1. The median value of the KHEI was 64.5, and participants were classified into low- and high-KHEI groups accordingly. Participants in the high-KHEI group were older and included a higher proportion of women, non-smokers, non-drinkers, regular exercisers, non-obese individuals, and those with hypertension. Furthermore, this group exhibited lower FPG and total cholesterol levels. In terms of kidney function, participants in the high-KHEI group exhibited lower eGFR. Conversely, UACR tended to be lower in the high-KHEI group.

### 3.2. The Risk of Prevalent CKD According to the KHEI

Although participants in the high-KHEI group had a lower eGFR, they had a lower prevalence of CKD (24.6%) than those in the low-KHEI group (28.0%). In addition, the risk of prevalent CKD was significantly lower in the high-KHEI group (adjusted odds ratio [aOR], 0.73; 95% confidence interval [CI], 0.58, 0.93) after adjustment with age, sex, and other covariates (Table 2). Furthermore, a higher KHEI, which was analyzed as a continuous variable, was significantly associated with a reduced risk of prevalent CKD, with an aOR of 0.98 (95% CI, 0.97, 0.99) per 1-point increment in the KHEI, as visualized in Figure 2 using a restricted cubic spline curve.

### 3.3. Subgroup Analysis

Participants with a higher KHEI exhibited a lower risk of prevalent CKD than those with a lower KHEI in all subgroups. Among several factors, the presence of hypertension (interaction *p* = 0.041) and educational level (interaction *p* = 0.044) were significantly associated with the risk of prevalent CKD according to the KHEI status (Table 3). The association between a higher KHEI and a lower risk of CKD was significant only in patients without hypertension (aOR, 0.57 [0.37, 0.87]) and those with a higher educational level (aOR, 0.56 [0.38, 0.81]). Although other factors did not exhibit significant interactions, the associations of higher KHEI and lower risk of prevalent CKD were prominent among participants aged <65 years, men, those with higher income levels, those with household members, residents in rural areas, and married participants. Similar results were observed in the subgroup analysis conducted with the KHEI as a continuous variable (Appendix A).

### 3.4. Healthy Dietary Patterns According to the Presence of CKD

In participants with CKD, the total KHEI was significantly lower than that in those without CKD when the mean values of the total score were fully adjusted (Appendix A). When comparing each component of the KHEI, the adjusted mean scores corresponding to the moderation and balance categories exhibited no significant differences regardless of the presence of CKD. In contrast, the adjusted mean scores for breakfast consumption, total fruit intake, fresh fruit intake, and milk and dairy product intake, which were included in the adequacy category, were significantly lower in the participants with CKD than in those without CKD (Figure 3).

In the subgroup analysis, participants without CKD had a higher KHEI across most subgroups than those with CKD. These findings were particularly prominent in participants aged <65 years, those without hypertension, those with a higher educational level, those with household members, those residing in rural areas, and married participants (Appendix A). Among the 14 KHEI components, the scores for total fruit intake and fresh fruit intake were significantly lower in those with CKD, particularly in the various subgroups of those who were older, men, those without hypertension, those with household members, those residing in rural areas, those with occupations, and married participants. From the perspective of subgroups, the significance of the differences in scores for each KHEI component according to the presence of CKD was most often found to vary by residential area.

## 4. Discussion

A healthy dietary pattern with a higher KHEI was significantly associated with a lower risk of prevalent CKD in patients with diabetes. We also found that the association of a higher KHEI with a lower risk of prevalent CKD was particularly pronounced among those without other comorbidities, those with a higher SES, and those living in supportive social environments. In addition, patients with diabetes and CKD exhibited a significantly lower KHEI than those without CKD, especially in the components of the adequacy category, which evaluates sufficient intake of recommended foods. Therefore, our study suggests that clinicians should emphasize the significance of a healthy dietary pattern, particularly in patients with diabetes at risk of CKD.

To the best of our knowledge, our study is the first to explore the relationship between KHEI and prevalent CKD, with a particular focus on patients with diabetes. The KHEI was developed and tailored to evaluate the eating habits and nutritional status of the Korean population. Because insufficient energy intake is more prevalent than excessive energy intake in Korean adults, the KHEI includes a balance category that covers both total energy intake and the percentage of energy derived from carbohydrates or fats [19]. Considering that the balance category was not included in the HEI, our study offers a more culturally suitable insight into the Korean population. Although previous studies in other countries have also reported an inverse association between healthy dietary patterns and CKD risk, cross-country comparisons require careful interpretation due to differences in dietary indices [23,24]. The KHEI includes culturally specific components, such as breakfast consumption and carbohydrate-fat energy balance, which are not captured in commonly used indices like HEI or Dietary Approaches to Stop Hypertension (DASH). Moreover, the categorization of food groups also varies across indices; for instance, KHEI groups protein sources together, while other indices evaluate them separately. These differences highlight the importance of using culturally appropriate dietary assessment tools when evaluating CKD risk in specific populations. However, consistent with previous research involving different ethnic groups [15,16], our study underscores culturally tailored nutritional education for higher adherence to healthy dietary patterns in patients with diabetes, particularly considering the consistently low adherence rates observed within this population [25,26].

While our findings support the beneficial effects of healthy dietary patterns, it is also important to consider the direct physiological impact of specific food groups. Diets rich in whole foods, including fruits, vegetables, and legumes, have been associated with improved glycemic control and kidney function [27]. Nonetheless, the precise roles of individual components such as dietary fiber, antioxidants, and phytochemicals in CKD remain to be fully elucidated, warranting further mechanistic studies and biomarker-based investigations. Furthermore, we compared the scores of each KHEI component among patients with diabetes with or without CKD to identify the specific dietary factors that may increase the CKD risk. Among the three main KHEI categories, the components of dietary adequacy were significantly different between patients with and without CKD.

First, skipping breakfast, a specific component included only in the KHEI [19], was identified as a risk factor for CKD in patients with diabetes. Previously, it has been demonstrated that irregular meal times and prolonged fasting can impair the regulation of hormones, including leptin, insulin, and glucocorticoids [28,29]. Thus, skipping breakfast has been identified as a risk factor for various comorbidities such as hypertension, dyslipidemia [30,31,32], and proteinuria [28]. Although the direct impact of skipping breakfast on the incidence of CKD remains inconclusive, the established link between metabolic diseases, proteinuria, and CKD development suggests that this association may have significant implications.

Second, low fruit intake has been identified as a risk factor for CKD in patients with diabetes. A higher dietary acid load causes metabolic acidosis and leads to a higher risk of CKD progression [33,34,35]. Fruits and vegetables, which are known as base-forming foods, mitigate kidney injury and preserve kidney function by improving metabolic acidosis [36,37]. An increased acid load due to low fruit intake increases endothelin-1, which causes endothelial dysfunction and atherosclerosis, and thus, the development of CKD status [38]. Although vegetables, along with fruits, reduce acid load, we could not find a significant negative association between vegetable intake and CKD risk. Considering that Korea has the highest per person vegetable consumption globally [39], the protective effect of vegetable intake against CKD may be attenuated compared to that of fruit intake. However, since a diet rich in vegetables reduces the incidence of CKD [40], adequate vegetable intake may be necessary, irrespective of our results.

Lastly, a low intake of milk and dairy products was identified as a risk factor for CKD in patients with diabetes. A systematic review of prospective cohort studies found a protective effect of dairy intake, especially low-fat dairy products, against CKD incidence and progression [41]. A variety of components found in dairy products, such as bioactive peptides, calcium, magnesium, and medium-chain fatty acids, have been demonstrated to possess antihypertensive properties, which are achieved through the inhibition of angiotensin-converting enzymes [42,43]. Additionally, these components possess anti-oxidative and anti-inflammatory properties [42,44], which may contribute to the protection of kidney function.

Our subgroup analyses indicate that the association between higher KHEI scores and lower CKD risk is more pronounced in diabetic patients without hypertension, those with higher educational levels, and those residing in rural areas. These results could be related to better overall health management and access to nutrition information and resources, as demonstrated in previous studies investigating healthy literacy levels related to hypertension [45] and educational level [46,47]. However, the detrimental impact of comorbidities or lower socioeconomic and educational status on disease progression may outweigh the benefits of dietary control. Given that higher blood pressure and lower SES, specifically lower educational status, are closely associated with increased incidence, prevalence, and progression of CKD [48,49], clinicians should integrate these factors with diet quality when assessing the CKD risk.

Interestingly, our study demonstrated that a higher KHEI was significantly associated with a lower risk of prevalent CKD in participants residing in rural areas. This finding contrasts with previous research, which typically indicated that rural areas have less access to healthcare services, leading to poorer health outcomes [50,51]. However, in South Korea, where healthcare accessibility is relatively high across all regions, this result may be attributed to differences in dietary habits and other factors [52]. In this regard, traditional diets, which are more prevalent in rural areas, may align more closely with the KHEI components, contributing to a high-quality diet [53,54].

This study had several limitations. First, owing to the observational and cross-sectional nature of the study, a causal relationship between the KHEI and CKD could not be established, and residual confounding factors and reverse causation may be present. In particular, information on specific medication use, duration since diagnosis, and quality of disease management was not available in the KNHANES dataset. These unmeasured factors could have influenced the observed associations and may contribute to residual confounding. Second, the study was conducted using data from a single country. Therefore, further studies with diverse ethnic groups are necessary to expand the generalizability of the findings. Third, the KHEI was calculated based on the participant’s memory, which is inherently susceptible to recall bias. Fourth, measurement errors may arise from questionnaire-based assessments. Therefore, future research should aim to identify objective biomarkers of food intake to complement existing dietary indices. Fifth, multiple testing across KHEI components and subgroups may have increased the risk of type I errors.

## 5. Conclusions

A healthy dietary pattern—particularly in the dietary adequacy category, which includes breakfast, fruit intake, and the consumption of milk and dairy products—may have a protective effect against CKD risk in Korean patients with diabetes. Consequently, clinicians should provide comprehensive dietary guidance to patients with diabetes who are at a high risk of developing CKD. Further research is warranted to investigate the effects of dietary interventions based on the findings of the present study.

## Figures and Tables

**Figure 1 nutrients-17-01600-f001:**
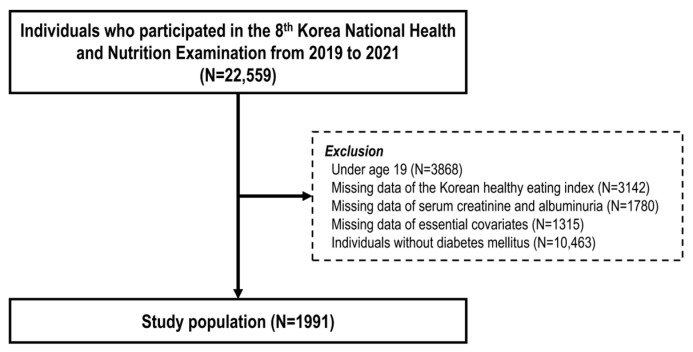
Study flow diagram.

**Figure 2 nutrients-17-01600-f002:**
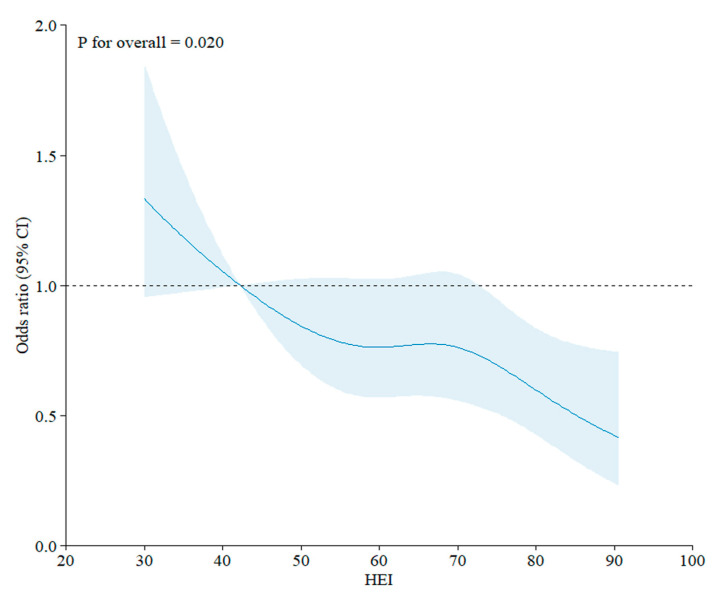
Restricted cubic spline curve demonstrating the association between the KHEI and risk of prevalent CKD. The *X*-axis indicates the total KHEI. The *Y*-axis indicates the odds ratio for the risk of prevalent CKD adjusted for age, sex, BMI, history of hypertension and dyslipidemia, education level, income status, occupation status, smoking, alcohol consumption, and physical activity. The area shaded in light blue indicates the 95% confidence interval.

**Figure 3 nutrients-17-01600-f003:**
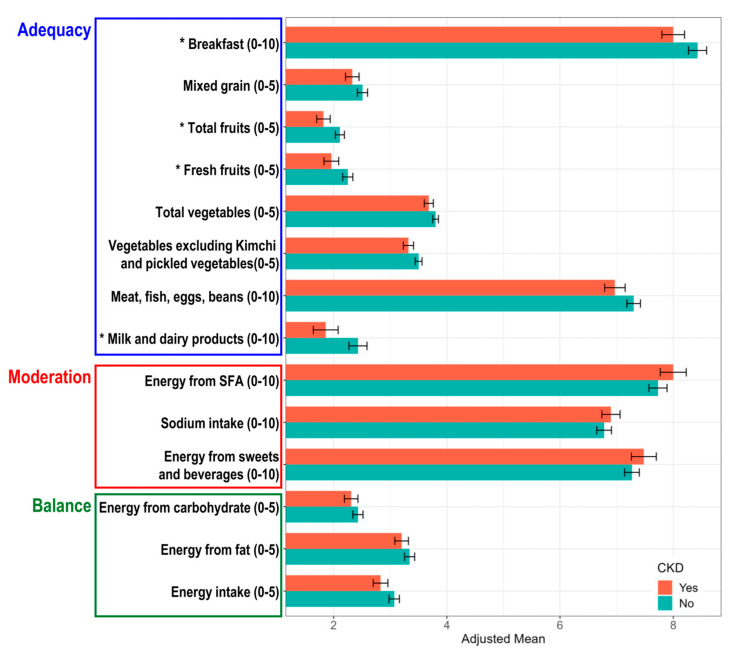
Comparison of each KHEI component according to the presence of CKD. The *X*-axis represents the adjusted mean scores of each KHEI component, controlling for age, sex, BMI, history of hypertension and dyslipidemia, education level, income status, occupation status, smoking, alcohol consumption, and physical activity. The *Y*-axis lists the KHEI components, with their respective minimum and maximum possible scores indicated in parentheses. Black lines on top of each bar represent 95% confidence intervals. Asterisks (*) denote statistically significant differences (*p* < 0.05).

**Table 1 nutrients-17-01600-t001:** Baseline characteristics of the study population according to the Korean Healthy Eating Index.

Variable	Korean Healthy Eating Index (KHEI)		*p*-Value
	Low (*N* = 995)	High (*N* = 996)	
Age, years old	58.4 ± 0.5	64.0 ± 0.5	<0.001
Age group			<0.001
<40 yr	82 (8.2%)	27 (2.7%)	
40–64 yr	587 (59%)	468 (47%)	
≥65 yr	326 (32.8%)	501 (50.3%)	
Sex, men	584 (58.7%)	522 (52.4%)	0.016
Education level			0.11
≤Elementary school	272 (27.3%)	306 (30.7%)	
Middle school	133 (13.4%)	164 (16.5%)	
High school	327 (32.9%)	290 (29.1%)	
>High school	263 (26.4%)	236 (23.7%)	
Income status, <20%	219 (22%)	189 (19%)	0.48
Number of household members, 1-person	169 (17%)	164 (16.5%)	0.019
Residence area			0.34
Urban	440 (44.2%)	415 (41.8%)	
Rural	555 (55.8%)	581 (58.3%)	
Occupation, yes	576 (57.9%)	516 (51.8%)	0.021
Marriage, yes	910 (91.5%)	953 (95.7%)	0.007
Smoking			<0.001
Non-smoker	461 (46.3%)	555 (55.7%)	
Ex-smoker	282 (28.3%)	284 (28.6%)	
Current smoker	252 (25.4%)	157 (15.8%)	
Alcohol			0.001
Non	322 (32.3%)	400 (40.2%)	
Mild to moderate	534 (53.7%)	514 (51.6%)	
Heavy	139 (14%)	82 (8.2%)	
Regular physical activity ^1^	321 (32.3%)	410 (41.2%)	0.001
Past medical history			
Hypertension	549 (55.2%)	602 (60.4%)	0.046
Dyslipidemia	461 (46.3%)	484 (48.6%)	0.38
Cardiovascular disease	113 (11.4%)	140 (14.1%)	0.14
BMI, kg/m^2^	26.4 ± 0.2	25.2 ± 0.1	<0.001
BMI group			<0.001
<18.5	11 (1.1%)	9 (0.9%)	
≥18.5 and <23	188 (18.9%)	248 (24.9%)	
≥23 and <25	201 (20.1%)	251 (25.2%)	
≥25 and <30	429 (43.1%)	398 (40%)	
≥30	166 (16.7%)	90 (9.1%)	
Waist circumference, cm	92.2 ± 0.4	89.6 ± 0.3	<0.001
Blood pressure, mmHg			
Systolic	125.0 ± 0.6	126.1 ± 0.6	0.16
Diastolic	77.1 ± 0.3	74.7 ± 0.4	<0.001
Laboratory measurements			
Fasting glucose, mg/dL	140.2 ± 1.7	133.8 ± 1.5	0.005
HbA1c, %	7.3 ± 0.1	7.2 ± 0.1	0.18
Total cholesterol, mg/dL	180.6 ± 1.7	168.8 ± 1.6	<0.001
HDL, mg/dL	46.2 ± 0.4	47.3 ± 0.4	0.06
Kidney function			
eGFR, mL/min/1.73 m^2^	87.8 ± 0.8	85.0 ± 0.8	0.012
eGFR < 60 mL/min/1.73 m^2^	80 (8%)	83 (8.3%)	0.82
UACR, mg/g	70.9 ± 9.7	43.8 ± 4.5	0.013
UACR group			0.040
<30 mg/g	759 (76.3%)	802 (80.5%)	
30–300 mg/g	190 (19.2%)	168 (16.9%)	
>300 mg/g	46 (4.6%)	26 (2.6%)	

Data are presented as the mean ± standard error for continuous variables or number (%) for categorical variables. BMI, body mass index; HDL, high-density lipoprotein; eGFR, estimated glomerular filtration rate; UACR, urine albumin–creatinine ratio. ^1^ Regular physical activity is defined as moderate-intensity physical activity ≥5 days or vigorous-intensity physical activity ≥3 days per week.

**Table 2 nutrients-17-01600-t002:** The risk of prevalent CKD according to the KHEI.

	Prevalence of CKD *, *N* (%)	Unadjusted		Age-Sex Adjusted		Model ^1^		Model ^2^	
		OR (95% CI)	*p*	OR (95% CI)	*p*	OR (95% CI)	*p*	OR (95% CI)	*p*
KHEI, as binary variable									
<Median value	279 (28.0)	1 (Reference)		1 (Reference)		1 (Reference)		1 (Reference)	
≥Median value	246 (24.6)	0.84 (0.67, 1.04)	0.12	0.68 (0.55, 0.84)	0.001	0.68 (0.54, 0.85)	0.001	0.73 (0.58, 0.93)	0.009
KHEI, as continuous variable	525 (26.4)	0.99 (0.98, 1.00)	0.06	0.98 (0.97, 0.99)	<0.001	0.98 (0.97, 0.99)	<0.001	0.98 (0.97, 0.99)	0.002

KHEI, Korean Healthy Eating Index; CKD, chronic kidney disease; OR, odds ratio; CI, confidence interval. * CKD is defined as eGFR < 60 mL/min/1.73 m^2^ or UACR ≥ 30 mg/g. ^1^ Multivariable model 1 is adjusted for age, sex, body mass index (BMI), and history of hypertension. ^2^ Multivariable model 2 is adjusted for age, sex, BMI, history of hypertension and dyslipidemia, education level, income status, occupation status, smoking, alcohol consumption, and physical activity.

**Table 3 nutrients-17-01600-t003:** Subgroup analysis of the prevalent CKD according to the KHEI.

	Subgroup	KHEI ^1^	*N*	Event (%)	aOR ^2^ (95% CI)	*p* *
Age	<65 years	Low	548	122 (23%)	1 (Reference)	0.12
		High	378	66 (16%)	0.64 (0.43, 0.95)	
	≥65 years	Low	447	167 (37%)	1 (Reference)	
		High	618	198 (33%)	0.84 (0.61, 1.15)	
Sex	Men	Low	520	157 (28%)	1 (Reference)	0.76
		High	480	126 (24%)	0.68 (0.48, 0.96)	
	Women	Low	475	132 (28%)	1 (Reference)	
		High	516	138 (26%)	0.78 (0.55, 1.11)	
Hypertension	No	Low	413	87 (21%)	1 (Reference)	0.04
		High	357	53 (13%)	0.57 (0.37, 0.87)	
	Yes	Low	582	202 (34%)	1 (Reference)	
		High	639	211 (33%)	0.80 (0.59, 1.08)	
Income status	<20%	Low	267	109 (44%)	1 (Reference)	0.83
		High	225	85 (39%)	0.75 (0.47, 1.18)	
	≥20%	Low	728	180 (23%)	1 (Reference)	
		High	771	179 (21%)	0.72 (0.54, 0.95)	
Education level	≤Middle school	Low	507	170 (32%)	1 (Reference)	0.04
		High	532	170 (32%)	0.98 (0.70, 1.35)	
	≥High school	Low	488	119 (25%)	1 (Reference)	
		High	464	94 (18%)	0.56 (0.38, 0.81)	
Number of households members	1-person	Low	211	77 (41%)	1 (Reference)	0.65
		High	186	62 (34%)	0.79 (0.48, 1.32)	
	≥2-person	Low	784	212 (25%)	1 (Reference)	
		High	810	202 (23%)	0.71 (0.54, 0.94)	
Residence area	Urban	Low	415	105 (26%)	1 (Reference)	0.17
		High	413	119 (28%)	0.92 (0.64, 1.33)	
	Rural	Low	580	184 (30%)	1 (Reference)	
		High	583	145 (23%)	0.62 (0.46, 0.84)	
Occupation	No	Low	474	176 (36%)	1 (Reference)	0.95
		High	514	156 (30%)	0.74 (0.53, 1.04)	
	Yes	Low	521	113 (23%)	1 (Reference)	
		High	482	108 (20%)	0.70 (0.49, 1.01)	
Marriage	No	Low	63	20 (36%)	1 (Reference)	0.36
		High	29	9 (23%)	0.48 (0.12, 1.96)	
	Yes	Low	932	269 (27%)	1 (Reference)	
		High	967	255 (25%)	0.74 (0.58, 0.94)	

^1^ The low and high KHEI groups were divided based on the median value of the KHEI. ^2^ The aOR is adjusted for age, sex, body mass index, history of hypertension and dyslipidemia, education level, income status, occupational status, smoking, alcohol consumption, and physical activity. * Interaction *p*-value. aOR, adjusted odds ratio; CI, confidence interval; CKD, chronic kidney disease; KHEI, Korean Healthy Eating Index.

## Data Availability

This study used data from the Korea National Health and Nutrition Examination Survey. The dataset(s) supporting the conclusions of this article is(are) available at https://knhanes.kdca.go.kr/knhanes/main.do (accessed on 24 March 2024).

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
