# Peer review of "Association Between Healthy Dietary Patterns and Chronic Kidney Disease in Patients with Diabetes: Findings from Korean National Health and Nutrition Examination Survey 2019–2021"

_nutrients, 2025, doi:10.3390/nu17091600_

Round 1
Reviewer 1 Report
Comments and Suggestions for Authors
In this study, the authors investigated the association between healthy dietary patterns, as assessed by the Korean Healthy Eating Index (KHEI), and the risk of prevalent CKD in patients with diabetes and identify differences in dietary patterns between patients with diabetes with and without CKD. The authors concluded that a healthy dietary pattern, particularly in the dietary adequacy category may have a protective effect against CKD risk in Korean patients with diabetes.
Comments:
The reviewer has some concerns as follows:
- In the Title, the word for “Title” is redundant and can be deleted.
- KHEI is the important key value for this study. The median values of the KHEI for low and high KHEI groups can be shown in the formal text, although there is information in supplemental file.
- In Tables 1 and 2, CKD markers are confusing. In Table 1, the “eGFR < 60 mL/min/1.73m2” is only 8% and 8.3% for low and high KHEI and the “UACR > 30 mg/g” is 23.8% and 19.5% for low and high KHEI. In Table 2, it described “CKD is defined as eGFR <60 mL/min/1.73 m2 or UACR ≥30 mg/g”. Therefore, how to calculate the CKD prevalence?
- In Figure 3, just questioning are there statistical differences in these results for comparison of each KHEI component according to the presence of CKD?
- Although this study is based on Korean nationals, differences from similar studies in other countries can be discussed, such as Cao et al., 2025, Association between dietary patterns and chronic kidney disease in elderly patients with type 2 diabetes: a community-based cross-sectional study. Nutr J. 2025 Jan 3;24(1):1.; Huang et al., The Combined Effects of the Most Important Dietary Patterns on the Incidence and Prevalence of Chronic Renal Failure: Results from the US National Health and Nutrition Examination Survey and Mendelian Analyses. Nutrients. 2024 Jul 12;16(14):2248.
- Overall, this manuscript needs a revision before it can be accepted.
Author Response
Comment R1-1. In the Title, the word for “Title” is redundant and can be deleted.
Response R1-1. Thank you for your comments. According to your comment, we revised the title of our study to: “Association between Healthy Dietary Patterns and Chronic Kidney Disease in Patients with Diabetes: Findings from the Korean National Health and Nutrition Examination Survey 2019-2021".
Comment R1-2. KHEI is the important key value for this study. The median values of the KHEI for low and high KHEI groups can be shown in the formal text, although there is information in supplemental file.
Response R1-2. Thank you for your comments. According to your comment, we presented the median value of the KHEI for low and high KHEI groups in the ‘Results’ section of the revised manuscript.
=============================================
Results section, Page 4, Line 176-177
3.1. Baseline characteristics
The baseline characteristics of the study participants divided by the median KHEI values are presented in Table 1. The median value of the KHEI was 64.5, and participants were classified into low- and high-KHEI groups accordingly.
Comment R1-3. In Tables 1 and 2, CKD markers are confusing. In Table 1, the “eGFR < 60 mL/min/1.73m2” is only 8% and 8.3% for low and high KHEI and the “UACR > 30 mg/g” is 23.8% and 19.5% for low and high KHEI. In Table 2, it described “CKD is defined as eGFR <60 mL/min/1.73 m2 or UACR ≥30 mg/g”. Therefore, how to calculate the CKD prevalence?
Response R1-3. Thank you for your comments. As we mentioned in ‘Materials and Methods’ section, we defined CKD as an estimated glomerular filtration rate (eGFR) <60 mL/min/1.73 m2 or urine albumin-creatinine ratio (UACR) ≥30 mg/g. In table 1, while the prevalence of eGFR <60 mL/min/1.73m2 was similar between the low and high KHEI groups (8.0% vs. 8.3%), the proportion of individuals with UACR ≥30 mg/g was notably higher in the low KHEI group (23.8% vs 19.5%). Therefore, the overall prevalence of CKD—defined by either criterion—was higher in the low KHEI group, as shown in Table 2. We emphasized the inclusion of albuminuria in the CKD definition because it reflects kidney damage even in the presence of preserved eGFR. This approach utilizes the strengths of the Korean National Health and Nutrition Examination Survey (KNHANES), which provides reliable, nationwide data on albuminuria based on quantitative measurements.
Comment R1-4. In Figure 3, just questioning are there statistical differences in these results for comparison of each KHEI component according to the presence of CKD?
Response R1-4. Thank you for your comments. We presented a comparison of the total and individual component scores according to the presence of CKD in Table S3. Please note that we organized and illustrated the results of multivariable model 2 from Table S3 in Figure 3. In response to your suggestion, we added asterisks (*) in Figure 3 to indicate components with statistically significant differences. The figure legend has also been updated accordingly to explain the meaning of the asterisks as follow.
=============================================
Figure 3. Comparison of each KHEI component according to the presence of CKD.
The X-axis represents the adjusted mean scores of each KHEI component, controlling for age, sex, BMI, history of hypertension and dyslipidemia, education level, income status, occupation status, smoking, alcohol consumption, and physical activity. The Y-axis lists the KHEI components, with their respective minimum and maximum possible scores indicated in parentheses. Black lines on top of each bar represent 95% confidence intervals. Asterisks (*) denote statistically significant differences (p < 0.05).
Comment R1-5. Although this study is based on Korean nationals, differences from similar studies in other countries can be discussed, such as Cao et al., 2025, Association between dietary patterns and chronic kidney disease in elderly patients with type 2 diabetes: a community-based cross-sectional study. Nutr J. 2025 Jan 3;24(1):1.; Huang et al., The Combined Effects of the Most Important Dietary Patterns on the Incidence and Prevalence of Chronic Renal Failure: Results from the US National Health and Nutrition Examination Survey and Mendelian Analyses. Nutrients. 2024 Jul 12;16(14):2248. Overall, this manuscript needs a revision before it can be accepted.
Response R1-5. Thank you for your insightful comment. Following your suggestions, we have discussed the findings of similar studies conducted in other countries to provide broader contest and support for our results. While prior research in both Western and East Asian populations has reported associations between healthy dietary patterns and reduced CKD risk, we emphasized that the structure and components of dietary indices differ across cultures. The KHEI uniquely reflects Korean dietary characteristics, such as breakfast consumption and carbohydrate–fat energy balance, which are not captured in commonly used indices like HEI or DASH. These differences were discussed to highlight the importance of culturally tailored dietary assessments when evaluating CKD risk. These points have been incorporated into the revised manuscript and are described in the ‘Discussion’ section as follows.
=============================================
Discussion section, Page 9, Line 277-286
Although previous studies in other countries have also reported an inverse association between healthy dietary patterns and CKD risk, cross-country comparisons require careful interpretation due to differences in dietary indices. The KHEI includes culturally specific components, such as breakfast consumption and carbohydrate-fat energy balance, which are not captured in commonly used indices like HEI or Dietary Approaches to Stop Hypertension (DASH). Moreover, the categorization of food groups also varies across indices; for instance, KHEI groups protein sources together, while other indices evaluate them separately. These differences highlight the importance of using culturally appropriate dietary assessment tools when evaluating CKD risk in specific populations.

Reviewer 2 Report
Comments and Suggestions for Authors
The manuscript submitted by Kim et al., titled: "Title Association between Healthy Dietary Patterns and Chronic Kidney Disease in Patients with Diabetes: Findings from the Korean National Health and Nutrition Examination Survey 2019-2021" is an interesting article describing a human study aiming to investigate the association between healthy dietary patterns and CKD in patients with diabetes from a Korean cohort. The manuscript is well written and discusses an important issue with clinical implications.
The reviewer would like to offer below a few points for the benefit of the authors and place these points under their consideration for the improvement of the manuscript:
- BMI does not have units it is an index and it is calculated not measured.
- Does the albumin:creatinine ratio need units? As a ratio it is an indication of the fold of one presence over the other in blood.
- Consider providing statistics on prevalence and cost/burden of diabetes in Korea, in the introduction section.
- Also consider including relevant statistics per CKD secondary to diabetes in Korea.
- In the methods section provide a reference for the KNHANES and the major eligibility criteria (inclusion and exclusion) for the KNHANES study.
- Please consider phrasing the selection mode of the study population more clearly. For example were all the participants who were eligible based on the eligibility criteria mentioned selected out of the KNHANES?
- It is inferred from the text that the KHEI is validated for the population used with but it would be important for that to be stated clearly accompanied with the appropriate reference.
- How did the authors consider and normalize for the following potentially confounding factors: smoking, physical activity, medication, years since diagnosis, and quality of management of the diseases?
Author Response
Comment R2-1. BMI does not have units it is an index and it is calculated not measured.
Response R2-1. Thank you for your comments. We agree that BMI is a calculated index rather than a directly measured value. In our revised manuscript, we have clarified this by explicitly stating in the Methods section that BMI was calculated as weight in kilograms divided by the square of height in meters (kg/m²). Although BMI is a calculated index, it is conventionally expressed with units of kg/m2 to reflect its derivation from measurable quantities and to aid in clinical interpretation. We have retained the unit notation in accordance with standard scientific reporting practices to ensure clarity and consistency.
=============================================
Materials and Methods section, Page 3, Line 143-145
2.4. Data collection and measurements
Anthropometric measurements including height, weight, waist circumference, and systolic and diastolic blood pressure were obtained. Body mass index (BMI) was calculated as weight (kg) divided by height squared (m²), and expressed in kg/m².
Comment R2-2. Does the albumin:creatinine ratio need units? As a ratio it is an indication of the fold of one presence over the other in blood.
Response R2-2. Thank you for your comments. Please kindly note that the urinary albumin-to-creatinine ratio (UACR) is conventionally expressed in the unit of mg/g, as specified in the KDIGO 2024 Clinical Practice Guideline for the Evaluation and Management of Chronic Kidney Disease [1]. As we adopted the definition of CKD in 2024 KDIGO guideline, we employed the same unit of UACR in this study.
[1] Kidney Disease: Improving Global Outcomes (KDIGO) CKD Work Group. KDIGO 2024 Clinical Practice Guideline for the Evaluation and Management of Chronic Kidney Disease. Kidney Int. 2024 Apr;105(4S):S117-S314.
Comment R2-3. Consider providing statistics on prevalence and cost/burden of diabetes in Korea, in the introduction section.
Response R2-3. Thank you for your comments. Based on your comment, we provided the prevalence and economic burden of diabetes in Korea in the ‘Introduction’ section as follows.
=============================================
Introduction section, Page 2, Line 55-57
Diabetes mellitus is a significant public health concern in Korea, with a prevalence of 12.5% among adults. The medical expenditures pertaining to diabetes have exhibited consistent upward trend, reaching a total medical cost of 2.5 billion United States dollar in 2022.
Comment R2-4. Also consider including relevant statistics per CKD secondary to diabetes in Korea.
Response R2-4. Thank you for your comments. Based on your comment, we provided the prevalence of diabetic kidney disease among patients with diabetes in the ‘Introduction’ section as follows.
=============================================
Introduction section, Page 2, Line 59-60
In Korea, 25.4% of patients diagnosed with DM also have diabetic kidney disease.
Comment R2-5. In the methods section provide a reference for the KNHANES and the major eligibility criteria (inclusion and exclusion) for the KNHANES study.
Response R2-5. Thank you for your valuable comments. As indicated in the revised manuscript, we have already provided a citation for the Korean National Health and Nutrition Examination Survey (KNHANES) with the reference number 21 in the revised manuscript [1]. In response to your suggestion, we have added a description of the sampling methodology and major eligibility criteria for the KNHANES population in the “Materials and Methods” section as follows.
=============================================
Materials and Methods section, Page 2, Line 93-98
A two-stage stratified cluster sampling method was employed, with districts and households as the primary and secondary sampling units, respectively. Within each sampled district, a random selection of households was made, with the exclusion of collective living facilities (e.g., nursing homes, military bases, or prisons) and foreign households. The survey was conducted with all household members aged one year or older.
[1] Kweon S, Kim Y, Jang MJ, Kim Y, Kim K, Choi S, Chun C, Khang YH, Oh K. Data resource profile: the Korea National Health and Nutrition Examination Survey (KNHANES). Int J Epidemiol. 2014 Feb;43(1):69-77.
Comment R2-6. Please consider phrasing the selection mode of the study population more clearly. For example were all the participants who were eligible based on the eligibility criteria mentioned selected out of the KNHANES?
Response R2-6. Thank you for your comments. Following our response to R2-5, we would like to clarify that we initially included all participants from the eighth KNHANES, which was conducted between 2019 and 2021, in the study population. Please kindly note that we already described detailed exclusion criteria and numbers of excluded participants in the ‘Materials and Methods’ section and Figure 1. Based on your comment, we revised Figure 1 in order to present this process more clearly.
Comment R2-7. It is inferred from the text that the KHEI is validated for the population used with but it would be important for that to be stated clearly accompanied with the appropriate reference.
Response R2-7. Thank you for your comments. We apologize for the omission of information regarding the validity of the KHEI. Please kindly note that KHEI was developed to evaluate the overall diet quality of Korean adults based on the National Dietary Guidelines for Koreans and the 2015 Dietary Reference Intakes for Koreans [1], although further validation studies are warranted. Moreover, please note that KHEI was developed to be calculated based on KNHANES data, an existing epidemiological resource that enables the analysis of dietary factors associated with chronic diseases. Thus, researchers commonly employ the KHEI when evaluating diet quality using KNHANES data. This has been specified in the ‘Materials and Methods’ section of the revised manuscript.
=============================================
Materials and Methods section, Page 3, Line 127-131
Although further validation studies of the KHEI are warranted, it is commonly used in research utilizing KNHANES data, as the KHEI was developed to evaluate the overall diet quality of Korean adults based on the National Dietary Guidelines for Koreans and the 2015 Dietary Reference Intakes for Koreans.
[1] Shin S, Kim S, Joung H. Evidence-based approaches for establishing the 2015 Dietary Reference Intakes for Koreans. Nutr Res Pract. 2018 Dec;12(6):459-468.
Comment R2-8. How did the authors consider and normalize for the following potentially confounding factors: smoking, physical activity, medication, years since diagnosis, and quality of management of the diseases?
Response R2-8. Thank you for your insightful comments. We included smoking status and physical activity as covariates in our multivariable models to account for their potential confounding effects. Smoking status was classified as non-smoker, ex-smoker, or current smoker, as specified in the ‘Materials and Methods’ (page 3, paragraph 4, lines 137-138). Physical activity was defined as a binary variable, indicating whether participants engaged in regular physical activity (≥20 minutes of vigorous-intensity activity on ≥3 days/week or ≥30 minutes of moderate-intensity activity ≥ 5 days/week), as specified in our manuscript.
As both variables were used categorically in regression models, statistical normalization was not required. Unfortunately, information regarding specific medication, duration since diagnosis, and quality of disease management was not available in the KNHANES dataset and therefore could not be considered in the analysis. We have acknowledged this limitation in the revised manuscript in discussion section.
=============================================
Materials and Methods section, Page 3, Line 137-138
Smoking status was classified as non-smoker, ex-smoker, or current smoker.
Discussion section, Page 11, Line 352-356
This study had several limitations. First, owing to the observational and cross-sectional nature of the study, a causal relationship between the KHEI and CKD could not be established, and residual confounding factors and reverse causation may be present. In particular, information on specific medication use, duration since diagnosis, and quality of disease management was not available in the KNHANES dataset. These unmeasured factors could have influenced the observed associations and may contribute to residual confounding.

Reviewer 3 Report
Comments and Suggestions for Authors
The authors conducted a cross-sectional study by using data from KNHANES to evaluate the relationship between healthy eating habits and the risk of chronic kidney disease (CKD) in diabetic individuals. CKD is a major global health issue and has a great impact on patients with diabetes, who often experience renal consequences. The connection between dietary patterns and the advancement of CKD shows how eating practices might lessen negative consequences in these individuals: a diet reduced in protein and sodium can help control hypertension and lessen the strain on the kidneys (10.1056/NEJMra1700312).Authors should underline how a diet high in whole foods has been shown to have a direct effect on blood glucose levels and kidney health in general (10.1136/bmj.k2396) and consider further investigations with specific biomarkers. Furthermore, the function of particular nutrients, like fiber, antioxidants, and phytochemicals, in the treatment of chronic kidney disease is still poorly understood and should be explored.
Author Response
Comment R3-1. Authors should underline how a diet high in whole foods has been shown to have a direct effect on blood glucose levels and kidney health in general (10.1136/bmj.k2396) and consider further investigations with specific biomarkers. Furthermore, the function of particular nutrients, like fiber, antioxidants, and phytochemicals, in the treatment of chronic kidney disease is still poorly understood and should be explored.
Response R3-1. Thank you for your valuable comments. Based on your comment, we highlighted the direct advantage of whole food-based diets on blood glucose regulation and kidney function, supported by previous findings [BMJ 2018;361:k2396]. We also emphasized the need for further investigation into the roles of individual nutrients—such as dietary fiber, antioxidants, and phytochemicals—whose specific effects on CKD remain insufficiently understood. Moreover, we revised the limitations section to reflect the necessity of identifying objective biomarkers for dietary intake to enhance the accuracy of pattern-based assessments. These revisions are included in the revised manuscript as follows.
=============================================
Discussion section, Page 9, Line 291-297
While our findings support the beneficial effects of healthy dietary patterns, it is also important to consider the direct physiological impact of specific food groups. Diets rich in whole foods, including fruits, vegetables, and legumes, have been associated with improved glycemic control and kidney function. Nonetheless, the precise roles of individual components such as dietary fiber, antioxidants, and phytochemicals in CKD remain to be fully elucidated, warranting further mechanistic studies and biomarker-based investigations.
Discussion section, Page 11, Line 359-361
Fourth, measurement errors may arise from questionnaire-based assessments. Therefore, future research should aim to identify objective biomarkers of food intake to complement existing dietary indices. Fifth, multiple testing across KHEI components and subgroups may have increased the risk of type I error.

Round 2
Reviewer 1 Report
Comments and Suggestions for Authors
This revised manuscript has a great improvement and the reviewer has no further comments.
Reviewer 2 Report
Comments and Suggestions for Authors
The authors have made a reasonable effort in addressing the reviewer's points.